# Phenotypic Assessment of Pathogenic Variants in *GNAO1* and Response to Caffeine in *C. elegans* Models of the Disease

**DOI:** 10.3390/genes14020319

**Published:** 2023-01-26

**Authors:** Martina Di Rocco, Serena Galosi, Francesca C. Follo, Enrico Lanza, Viola Folli, Alberto Martire, Vincenzo Leuzzi, Simone Martinelli

**Affiliations:** 1Department of Oncology and Molecular Medicine, Istituto Superiore di Sanità, 00161 Rome, Italy; 2Department of Human Neuroscience, ‘Sapienza’ University of Rome, 00185 Rome, Italy; 3Center for Life Nano Science, Istituto Italiano di Tecnologia, 00161 Rome, Italy; 4D-tails s.r.l., 00165 Rome, Italy; 5National Center for Drug Research and Evaluation, Istituto Superiore di Sanità, 00161 Rome, Italy

**Keywords:** *GNAO1*, Gαo, movement disorders, epilepsy, caffeine, *Caenorhabditis elegans*

## Abstract

De novo mutations affecting the G protein α o subunit (Gαo)-encoding gene (*GNAO1*) cause childhood-onset developmental delay, hyperkinetic movement disorders, and epilepsy. Recently, we established *Caenorhabditis elegans* as an informative experimental model for deciphering pathogenic mechanisms associated with *GNAO1* defects and identifying new therapies. In this study, we generated two additional gene-edited strains that harbor pathogenic variants which affect residues Glu^246^ and Arg^209^—two mutational hotspots in Gαo. In line with previous findings, biallelic changes displayed a variable hypomorphic effect on Gαo-mediated signaling that led to the excessive release of neurotransmitters by different classes of neurons, which, in turn, caused hyperactive egg laying and locomotion. Of note, heterozygous variants showed a cell-specific dominant-negative behavior, which was strictly dependent on the affected residue. As with previously generated mutants (S47G and A221D), caffeine was effective in attenuating the hyperkinetic behavior of R209H and E246K animals, indicating that its efficacy is mutation-independent. Conversely, istradefylline, a selective adenosine A_2A_ receptor antagonist, was effective in R209H animals but not in E246K worms, suggesting that caffeine acts through both adenosine receptor-dependent and receptor-independent mechanisms. Overall, our findings provide new insights into disease mechanisms and further support the potential efficacy of caffeine in controlling dyskinesia associated with pathogenic *GNAO1* mutations.

## 1. Introduction

*GNAO1* encodes the G protein α o subunit (Gαo), which is one of the most abundant proteins in the mammalian brain whose function is highly conserved throughout evolution [1,2,3]. Gαo controls inhibitory signaling from several G-protein-coupled receptors (GPCRs), which modulates neuronal excitability [4] and controls neurodevelopment [5,6]. In striatal medium spiny neurons, Gαo modulates signal flow through dopamine D_2_ and adenosine A_2A_ GPCRs and the second messenger cyclic adenosine monophosphate (cAMP) cascade and has major effects on motor control [7].

*GNAO1* pathogenic variants occur de novo in infantile- and childhood-onset neurological disorders. The phenotypic spectrum is highly heterogeneous and ranges from early infantile epileptic encephalopathy (EIEE17, MIM #615473) to neurodevelopmental disorder with involuntary movements, with or without epileptic seizures (NEDIM, MIM #617493) [8,9,10,11,12]. Susceptibility to a broad range of triggers, including emotions, fever, high external temperature, infections, and intentional movements causes life-threatening paroxysmal exacerbations and is pathognomonic for GNAO1 encephalopathy and other postsynaptic disorders caused by mutations in genes with a role in this pathway (i.e., *ADCY5*, *GNB1*, *HPCA*, *PDE2A*, and *PDE10A*) [13,14]. Atypical presentations associated with milder and delayed-onset dystonia have recently been reported [15,16].

Recent data from our study and others suggest that *GNAO1* mutations act through a combination of loss-of-function (LOF) and dominant-negative (DN) mechanisms, depending on the particular residue involved and the specific amino acid substitution [7,17,18,19]. However, a comprehensive understanding of the genotype/phenotype correlations and the underlying pathogenic mechanisms is still lacking, which hinders the discovery of effective therapies. In the winding path toward the identification of new potential drugs, simple model organisms can help. Recently, we have shown that caffeine dramatically improves the aberrant motor function of engineered nematodes that carry mutations in *goa-1*, the *C. elegans* orthologues of *GNAO1*, by blocking a putative adenosine receptor (AR) in the worm [18]. In the same year, Larasati and colleagues showed that dietary zinc supplementation restores both the motor function and the longevity of humanized flies harboring disease-causing *GNAO1* variants [19].

Here, we extended *C. elegans* studies to two of the most common *GNAO1* mutations associated with hyperkinetic movement disorder (MD), i.e., R209H and E246K. Functional assessment of monoallelic and biallelic variations revealed a variable LOF behavior of changes in Gαo-mediated signaling and a cell-specific DN effect, which was limited to the heterozygous R209H allele. Of note, caffeine was found to ameliorate hyperactive locomotion of gene-edited animals both in an AR-dependent and AR-independent manner.

## 2. Materials and Methods

### 2.1. Generation of Gene-Edited C. elegans Strains

Culture, maintenance, microinjections, and crosses were performed using standard techniques [20,21]. The Bristol N2 strain (control animals) was provided by the *Caenorhabditis* Genetics Center (CGC; University of Minnesota, Minneapolis, MN, USA).

Nucleotide changes to the endogenous *goa-1 locus* were carried out by CRISPR/Cas9 genome editing as previously described [18] but with minor modifications. Two single guides (sgRNAs) were assembled with the Cas9 protein to generate multiple ribonucleoprotein (RNP) complexes targeting codon 209. This strategy has recently been described to target multiple *C. elegans loci* [22]. Briefly, twenty N2 animals were injected with a mix that contained 750 ng/μL Cas9 (IDT, Coralville, Iowa, USA), 700 ng/μL ALT-R CRISPR tracrRNA, 115 ng/μL *dpy-10* crRNA, 37.5 ng/μL ssODN *dpy-10*, 350 ng/μL *goa-1* crRNA (5′-TACAGATTGTTCGATGTGGG-3′), 175 ng/μL ssODN *goa-1*[R209C] (5′-TCTTGATAAAAAAATAATATTTAATAATGAATATTTACAGATTATTTGACGTTGGTGGACAGAGATCTGAGTGTAAAAAATGGATTCATTGTTTCGAAGATGTTACTGCTATTAT-3′), and 175 ng/μL ssODN *goa-1*[R209H] (5′-TCTTGATAAAAAAATAATATTTAATAATGAATATTTACAGATTATTTGACGTTGGTGGACAGAGATCTGAGCATAAAAAATGGATTCATTGTTTCGAAGATGTTACTGCTATTAT-3′). Worms were then recovered on NGM medium at 20 °C. PCR amplification was carried out using a single forward primer (5′-TGACGACCTGGAAAGGTTAGG-3′) and two reverse primers annealing with wild-type (5′-CCACGAGGCATCTGCGTATAT-3′) or modified (5′-CGTTGGTGGACAGAGATCTG-3′) sequences to isolate animals harboring *goa-1* c.625_627AGG > TGT (R209C) or c.625_627AGG > CAT (R209H) nucleotide substitution. Animals were isolated from jackpot plates that corresponded to plates with a higher percentage of roller and dumpy worms, which was used as a co-injection phenotypic marker [23]. Genotype was confirmed by Sanger sequencing. Multiple independent lines were generated which carried the R209H substation but not the R209C substitution. Gene-edited lines were out-crossed three times to the N2 strain to remove any potential off-target mutation and were then used for phenotypic analysis. These lines displayed an equivalent phenotype and were referred to as *goa-1*(*pan23*[R209H]). We failed to identify gene-edited strains carrying the c.736G > A (E246K) variant (details on this microinjection are available upon request). The generation of the hT2-balanced strain PHX6121 *goa-1*(*syb5893*)/hT2[*bli-4*(*e937*)*let-?*(*q782*)qIs48] which carried this substitution in a heterozygous state was outsourced (Fujian SunyBiotech Co., Ltd., Fuzhou, China). From this strain, we then isolated animals homozygous for index mutation.

### 2.2. Sensitivity to Aldicarb, Levamisole and PTZ

Sensitivities to aldicarb, levamisole, and PTZ (Sigma-Aldrich, Saint Louis, MO, USA) were assessed as previously described [24,25,26]. Young adult hermaphrodites obtained from synchronized cultures were assayed. Paralysis was confirmed by evaluating the lack of response to prodding with a platinum wire. Convulsions were measured on agar plates. Phenotypic analyses were conducted using Leica MZ10F (Leica Microsystems, Wetzlar, Germany) and Nikon SMZ18 (Nikon Europe, Amstelveen, the Netherlands) stereomicroscopes. Data were analyzed as the percentage of animals that were still able to move at each time point.

### 2.3. Behavioral Assays

Locomotion parameters were analyzed quantitatively using a recently developed automated tracking system, which has been previously detailed [18]. Body bends are defined as a change in the direction of the posterior bulb part of the pharynx along the *y* axis when the worm is traveling along the *x* axis [27]. Reversals are defined as backward movements equal to at least one-fifth of the animal’s length that correspond to the length of the pharynx. Short reversals were defined as backward movements equal to at least ½ of the length of the pharynx. Synaptic vesicles release by HSN motor neurons was evaluated by counting the number of eggs retained in the uterus of adult hermaphrodites in the exponential phase of laying using an Eclipse Ti2-E microscope (Nikon Europe) which was equipped with DIC optics on live animals and mounted on 2% agarose pads that contained 10 mM sodium azide as anesthetic. Embryonic and larval lethality, Pvl phenotype, and brood size were investigated as previously reported [18].

### 2.4. Response to Caffeine and Istradefylline

Caffeine (Sigma-Aldrich) was freshly dissolved in water and added to agar plates (5 mM potassium phosphate buffer, pH 6.0, 1 mM CaCl_2_, 1 mM MgSO_4_) to a final concentration in the range from 1–10 mM. Plates were freshly prepared and seeded with 12.5 μL of *Escherichia coli* OP50 bacteria to get a thin and homogeneous lawn of food. The number of reversals per minute was counted in L3 animals that moved on the assay plate after 2 h of exposure. Sensitivity to istradefylline (Tocris, Bristol, UK) was assessed after 2 h of exposition at multiple doses. Istradefylline was freshly dissolved in DMSO. Given that DMSO alone affects reversal rates in gene-edited animals (0.3% and 0.6% DMSO decrease the reversal rates of 12% and 30%, respectively) and in control worms (0.3% and 0.6% DMSO decrease the reversal rates of 35 and 40%, respectively), nematodes were treated in parallel with the drug or the solvent alone at the corresponding dose. The percentage of rescue is defined as [1 − (Δ1/Δ2)*100], where Δ1 and Δ2 are the difference between the reversals measured in the presence and absence of the drug and the reversals of control worms exposed to the solvent, respectively. Therefore, the percentage of rescue was 0% when the reversal rate remained the same before and after drug exposition and 100% when the reversal rate resembled that of control animals after exposure to the drug.

### 2.5. Statistics

Statistical differences were calculated using GraphPad Prism 8.4.2 software. Speed distributions were evaluated by a two-sample t-test using MATLAB software version R2022a (Mathworks, Natick, Massachusetts). Genotype blinding was used for all experiments, except for the data reported in Figure 1.

## 3. Results

### 3.1. Generation of Gene-Edited C. elegans Strains

Human Gαo and *C. elegans* GOA-1 display 90% homology in their amino acid sequence, and the vast majority of affected residues, including the mutational hotspots Arg^209^ and Glu^246^, were conserved between the two species [17,18]. Initially, we aimed to introduce c.625_627AGG > TGT (p.R209C), c.625_627AGG > CAT (p.R209H), and c.736G > A (p.E246K) nucleotide substitutions at the orthologous location of the *C. elegans* gene by CRISPR/Cas9 genome editing. Due to the high efficiency of this technology in the worm, we used a multiplexed strategy by simultaneously expressing two sgRNAs to generate animals that carried both mutations at codon 209 in a single microinjection (Appendix A). However, while we obtained several independently edited animals, which harbored the R209H substitution, during multiple rounds of injections, we were unable to identify worms that carried the R209C change in either a homozygous or heterozygous state. Similarly, in independent sets of injections, we failed to identify modified animals that carried the glutamic acid-to-lysine amino acid substitution at codon 246. Given Gαo’s role in controlling mitotic spindle alignment during early embryogenesis [28,29] and regulating meiotic maturation of the germline [30], these results suggest a detrimental effect of specific *goa-1* mutations on embryonic survival and/or gametogenesis. Recent data by Wang and colleagues seem to confirm this hypothesis. In fact, although they succeeded in generating and characterizing R209C-edited animals [17], it was not possible to keep this strain in culture for a long time or freeze it due to the extremely low number of laid eggs [31]. A similar progeny issue was observed in worms homozygous for the G203R substitution. Based on these findings, we decided to focus our work on the arginine-to-histidine change at codon 209 and outsource the generation of an hT2-balanced strain that carried the c.736G > A (p.E246K) variant in heterozygosity (Fujian SunyBiotech Co., Ltd., Fuzhou, China). From this strain, we then isolated animals homozygous for the aforementioned mutation.

In sum, we generated *goa-1*(*pan23*[R209H]) animals (hereafter *goa-1*[R209H]) and derived homozygous E246K-edited worms from the balanced PHX6121 *goa-1*(*syb5893*)/hT2[*bli-4*(*e937*)*let-?*(*q782*)qIs48] strain (hereafter *goa-1*[E246K]). Like *goa-1*(*sa734*) null mutants [18,30,32] and previously generated knock-in animals [18], the nematodes, homozygous for the anticipated changes, displayed slow growth and variable percentages of nonspecific phenotypes, including protruding vulva (Pvl), larval and embryonic lethality, and reduced offspring (Table 1).

### 3.2. Biallelic GNAO1 Mutations Result in Loss of Gαo Function in C. elegans

A well-conserved function of Gαo and well-established behavioral phenotypes justified assessing the impact of *GNAO1* mutations in the worm. In *C. elegans*, Gαo signaling inhibits the synaptic vesicles release machinery in HSN and ventral cord motor neurons and controls egg laying and locomotion, respectively [33]. Consistently, Gαo null mutants display hyperactive egg laying and locomotion and show hypersensitivity to aldicarb, which is an acetylcholinesterase inhibitor that causes accumulation of acetylcholine (ACh) in the synaptic cleft of the neuromuscular junction (NMJ); in turn, this results in sustained muscle contraction and paralysis [33,34,35]. Previous studies established that gene-edited worms that harbor biallelic *goa-1* variants phenocopy the behavior of *goa-1*(*n363*) or *goa-1*(*sa734*) null mutants, which displays increased egg laying and locomotion and hypersensitivity to aldicarb-induced paralysis, suggesting augmented releases of neurotransmitters by both HSN and ventral cord motor neurons [17,18].

In this study, we show that homozygous R209H and E246K mutations affected the number of eggs retained in the uterus of adult hermaphrodites during the exponential phase of laying, with an average of 15 in the control worms to an average of 4 and 2.5 in *goa-1*[R209H] and *goa-1*[E246K] animals, respectively (Figure 1). These data suggest an excessive release of serotonin and other neurotransmitters by HSNs, which is likely due to a lack of Gαo-mediated inhibition in this pair of motor neurons.
Figure 1*goa-1*[R209H] and *goa-1*[E246K] animals show increased egg laying activity. (**A**) Representative images of mid-body regions of control animals (upper panel) and *goa-1* mutants (lower panels). Retained eggs are visible as oval objects inside the body of adult hermaphrodites. Wild-type worms displayed 15 unlaid eggs on average, while Gαo mutants retained only a few eggs in their uterus. Magnification is constant in all images. (**B**) The egg laying activity is quantified as the number of eggs present in the uterus (* *p* < 0.05 and *** *p* < 0.0001; one-way ANOVA with Bonferroni correction). Twenty animals for each genotype were tested. Data represent means ± SEM of multiple observations. Data collected in our previous study [18] are included in panel B for comparison (gray bars). *sa734* is a null allele of *goa-1* [30,32].
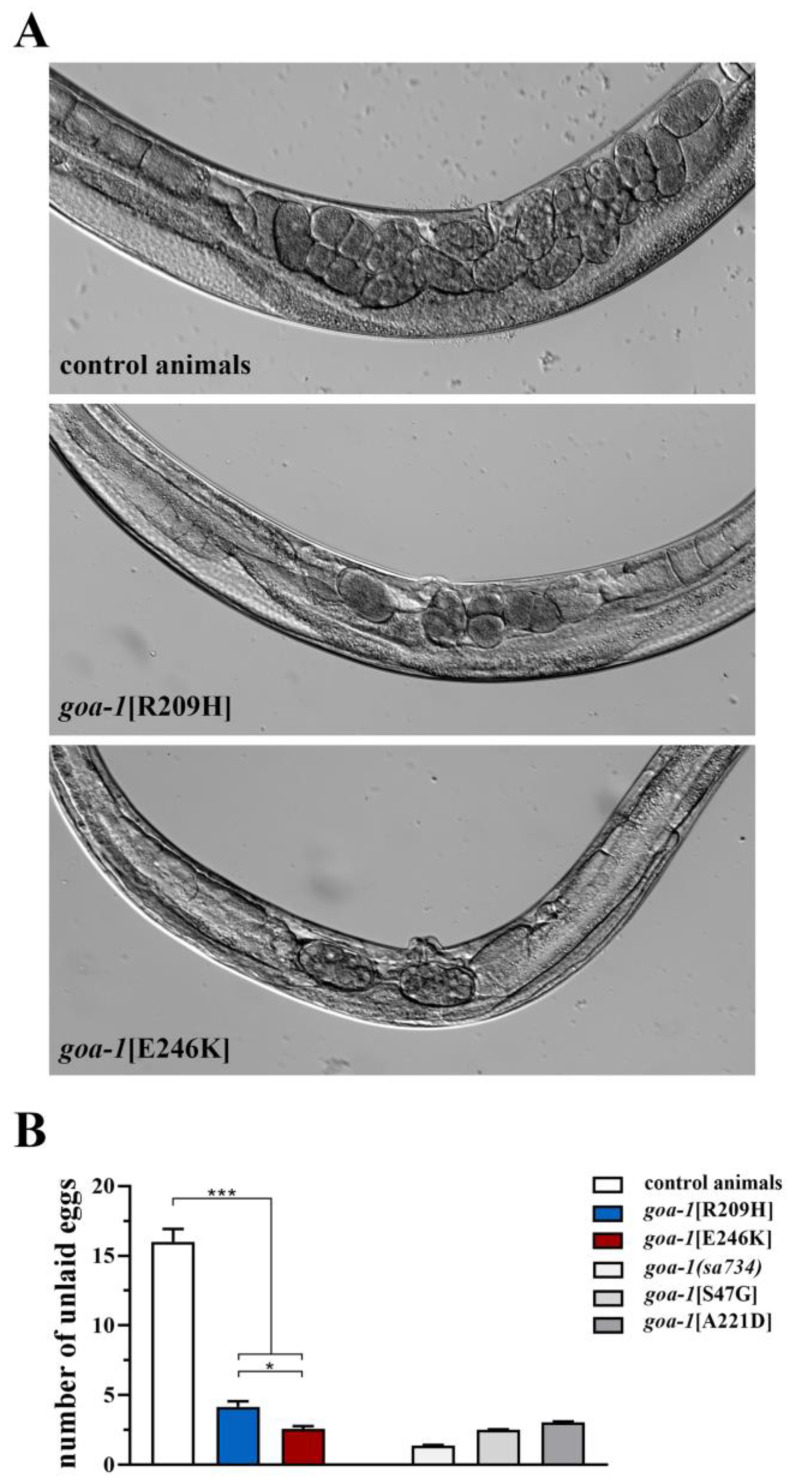



Similarly, *goa-1*[R209H] and *goa-1*[E246K] mutants phenocopied the hyperkinetic motor behavior of previously generated knock-in animals [17,18] and *goa-1* LOF mutants [17,18,33,34,35]. In particular, by using an automated tracking platform to record multiple locomotor parameters, we were able to demonstrate that *goa-1*[R209H] and *goa-1*[E246K] animals showed hyperactive spontaneous crawling (Figure 2A), moved faster than control animals (Figure 2B) with more frequent body bends (Figure 2C), and displayed uncoordinated locomotion, assuming a coil-like shape when they attempted to move (Figure 2D). Moreover, *goa-1*-edited worms exhibited an increased frequency of reversals (*i.e.*, spontaneous backward locomotion) (Figure 2E), likely due to the augmented release of glutamate onto AVA, which is the main interneuron that controls this repetitive behavior [36].

*C. elegans* locomotion is regulated by GPCR-GOA-1-mediated signaling, which negatively controls presynaptic ACh release by ventral cord motor neurons [37,38]. Pharmacological treatment with aldicarb represents a fast and reliable strategy to determine whether synaptic transmission at the NMJ is altered [24]. As anticipated, *goa-1* null mutants and previously generated CRISPR-edited worms which harbored *GNAO1* mutations showed hypersensitivity to aldicarb-induced paralysis because of augmented ACh release and its faster accumulation in the synaptic cleft [17,18]. Consistently, *goa-1*[R209H] and *goa-1*[E246K] animals exhibited variable hypersensitivity to this drug (Figure 3A), but normal sensitivity to levamisole (Appendix A), which is a nicotinic receptor agonist that acts on postsynaptic muscles, highlighting the presynaptic origin of the cholinergic defect. We then assessed GABAergic signaling at inhibitory motor neurons by evaluating sensitivity to pentylenetetrazole (PTZ), which is a competitive inhibitor of GABA-A receptors on body-wall muscles. Exposure to this drug resulted in a convulsive phenotype [25]. Of note, knock-in animals showed PTZ hypersensitivity (Figure 3B), indicating a shift in the equilibrium between excitatory and inhibitory inputs towards the former at the *C. elegans* NMJ. A particularly severe phenotype was observed in *goa-1*[E246K] mutants.

Overall, these findings further support the notion that pathogenic *GNAO1* mutations result in the loss of Gαo function when tested in *C. elegans* as homozygous alleles.

### 3.3. Heterozygous GNAO1 Mutations Display a Cell-Specific DN Behavior in C. elegans Neurons

Recent data indicate that *GNAO1* haploinsufficiency causes mild dystonic features associated with an age of onset in adulthood/adolescence and only minor neurological comorbidities [15,16]. This finding indicates that haploinsufficiency is not sufficient to cause the archetypal EIEE17 or NEDIM phenotypes, suggesting a more complex behavior for the mutant alleles underlying these conditions. Consistently, multiple lines of evidence indicated that pathogenic *GNAO1* mutations variably act dominantly by interfering with the function of wild-type Gαo [7,17,18].

In this study, we analyzed the functional consequences of heterozygous *goa-1* variants in the F1 progeny generated by genetic crosses between hermaphrodites that harbored biallelic mutations and males that carried the wild-type *goa-1* allele. Wang and colleagues have recently demonstrated that *goa-1*(+/−) animals behave as control worms in the aldicarb assay and show normal sensitivity to this drug [17]. Conversely, *goa-1*(+/R209H) displayed a hypersensitive response to aldicarb-induced paralysis, which was similar to that observed in the homozygous *goa-1*[R209H] animals (Figure 4A). Similarly, *goa-1*(+/R209H) nematodes exhibited an increased reversal rate, which resembled the phenotype observed in the homozygous strain (Figure 4B). In contrast, the hyperactive egg laying phenotype of *goa-1*[R209H] nematodes was partially restored by the presence of a single wild-type *goa-1* allele (Figure 4C). Finally, no relevant defects were observed in *goa-1*(+/E246K) heterozygous animals, at least for the analyzed phenotypes.

Taken together, these findings suggest that R209H acts as a DN allele in cholinergic ventral cord motor neurons mediating aldicarb sensitivity and locomotion and likely in glutamatergic AVA interneurons controlling the reversal rate, but not in serotoninergic HSN motor neurons, which regulate egg laying. Our data further demonstrate a cell-specific effect of individual *GNAO1* pathological changes in *C. elegans*.

### 3.4. Response to Caffeine and Istradefylline

Recently, we have shown that caffeine significantly improves the aberrant motor function of gene-edited worms harboring two pathogenic *GNAO1* mutations—S47G and A221D [18]. Such a benefic effect was mimicked by istradefylline, which is a selective adenosine A_2A_ receptor (A_2A_R) antagonist that is currently used in the treatment of Parkinson’s disease (PD) [39], and by other AR antagonists, indicating that caffeine acts, at least in part, by blocking a putative AR in the nematode [18].

In this study, two hours of exposure of *goa-1*[R209H] and *goa-1*[E246K] animals to caffeine restored hyperactive locomotion in terms of the number of short reversals per minute in a dose-dependent manner (Figure 5A). Interestingly, istradefylline was able to rescue the reversal rate of *goa-1*[R209H] worms but not *goa-1*[E246K] worms (Figure 5B and Appendix A), suggesting that caffeine acts through AR-dependent and AR-independent mechanisms. The reversal rate was not significantly affected in control animals that were exposed to these molecules.

## 4. Discussion

We report on the functional impact of two of the most recurrent *GNAO1* mutations associated with hyperkinetic MD. Like previously tested variants, when introduced in the *C. elegans* genome as homozygous alleles, these changes result in a loss of Gαo function in multiple types of neurons, causing hyperactive locomotion and egg laying. Phenotypic profiling of heterozygous variants suggests cell context-specific DN behavior of the R209H but not the E246K allele. Finally, we further confirm the beneficial effect of caffeine in restoring normal locomotion in gene-edited worms and establish that this molecule may act through AR-dependent and AR-independent mechanisms.

GNAO1 encephalopathy is characterized by a clinical phenotype combining distinctive motor, epileptic, and neurodevelopmental features [8,9,10,11,12]. While two different nosological entities are reported in OMIM (EIEE17 and NEDIM), a significant overlap in disease manifestations across patients carrying different variants best captures what is observed in clinical practice, with a small number of affected individuals presenting with isolated MD or epilepsy [12,13]. However, genotype-phenotype correlations have been reported as some variants have been consistently associated with MD, others with early onset epilepsy, and still others with severe MD with life-threatening exacerbations [12,13,14]. In addition, a subset of missense changes and LOF mutations that are predicted to result in *GNAO1* haploinsufficiency have recently been associated with milder and delayed-onset dystonia [15,16]. Based on this evidence, understanding the mechanisms by which individual mutations affect downstream signaling pathways is crucial for developing targeted therapies to treat this disorder effectively.

In this scenario, simple model organisms represent a powerful tool of investigation. In a humanized *Drosophila* model of GNAO1 encephalopathy, heterozygous flies carrying the G203R variant recapitulated some of the clinical features of the disease, manifesting motor dysfunction, reduced life span, and brain abnormalities [19,40]. Remarkably, these phenotypes were improved by dietary zinc supplementation [19]. In *C. elegans*, the generation of genetically modified strains established that biallelic *goa-1* variants (G42R, S47G, G203R, R209C, R209H, A221D, and E246K) show a clear LOF effect on Gαo-mediated signaling [17,18], but they affect diverse biological processes differently. For instance, only G203R, R209C, and E246K strongly affected developmental programs and the meiotic maturation of germ cells. Moreover, although E246K had no DN activity in our assays, *goa-1*[E246K] animals showed extreme severity of certain phenotypes (uncoordinated locomotion and PTZ-hypersensitivity), which were more severe than those of worms which lacked *goa-1*. Finally, *goa-1*[R209H] animals were very fast but less sensitive to aldicarb than other mutants, with only a slightly increased reversal rate. These differences likely reflect the unique impact of individual mutations on cellular and clinical phenotypes.

When assessed as monoallelic variants, some *GNAO1* mutations (G42R, R209C, R209H, A221D) functioned as DN alleles in nematodes, while others did not, at least in the experiments performed. Intriguingly, the A221D substitution had DN activity in HSN motor neurons, but not in ventral cord motor neurons. Similarly, the R209H allele acted dominantly in cholinergic neurons controlling locomotion and possibly in AVA interneurons but not in HSNs. These results suggest a neuron type-specific DN behavior of individual *GNAO1* alleles in *C. elegans*, which is in line with recent data obtained in neuronal cultures [7]. Specifically, these studies revealed that R209C and G203R mutations affected dopamine signaling in medium spiny neurons expressing D_2_ receptors (iMSNs), but only G203R increased dopamine response in MSNs expressing D_1_ receptors (dMSNs). Similarly, both changes impaired adenosine signaling in dMSNs, but only G203R improved the strength of adenosine responses in iMSNs. Overall, these findings suggest that *GNAO1* may affect motor control in a mutation-dependent manner.

Caffeine was shown to ameliorate the hyperactive MD of genetically modified worms carrying the p.S47G and p.A221D *goa-1* variants by antagonizing a putative AR [18]. In this study, we confirmed the beneficial effect of this molecule in *goa-1*[R209H] and *goa-1*[E246K] animals. Of note, the selective A_2A_R antagonist istradefylline effectively improved the reversal rate of *goa-1*[R209H] but not *goa-1*[E246K] worms. Such a discrepancy could be explained by the well-known propensity of caffeine to act through both AR-dependent and AR-independent mechanisms. In this regard, caffeine is also a nonspecific phosphodiesterase (PDE) blocker [41,42], while istradefylline does not induce the inhibition of canine PDE I-V enzymes [43]. In line with this finding, caffeine could be effective on phenotypes resulting from mutations that involve both increased and decreased cAMP levels, which block ARs or PDE, respectively; in contrast, istradefylline should reasonably act only when cAMP abnormally increases. The effect of Gαo mutations on intracellular cAMP levels has not yet been systematically characterized. Recent data from Larasati et al. revealed an increased association between Gαo-E246K and Gβγ subunits [19]. Since the release of Gβγ from the heterotrimeric complex is required to positively modulate the responsiveness of the cAMP-producing enzyme adenylyl cyclase 5 (AC5) [7], which is the main AC isoform in the striatum [44], we might expect decreased cAMP levels in *goa-1*[E246K] animals, which could explain the lack of response to istradefylline. The amplified inhibition of cAMP synthesis observed in HEK 293T cells overexpressing the E246K but not the R209H variant [45] appears to support this model. However, further studies are needed to assess the consequences of individual *GNAO1* variants on Gβγ signaling.

The therapeutic potential of caffeine has already been established in PD and other MDs. It is worth noting that an ongoing clinical trial (ClinicalTrials.gov; id.: NCT04469283) is evaluating the effects of caffeine in subjects with mutations in *ADCY5*, which is the gene coding for AC5 [46], and a retrospective study regarding the efficacy of caffeine in *ADCY5*-related dyskinesia has recently been published [47]. Despite limitations related to the rarity of the condition and the nonrandomized design, this study, which included 30 patients with *ADCY5*-related dyskinesia, showed that caffeine is well tolerated in children, and 27 patients reported a clear reduction in motor symptoms and a consistent improvement in their quality of life. As for the three patients who reported a worsening quality of life, the impact of the identified variants on the modulation of cAMP levels has not been investigated. We can only speculate that *ADCY5* mutations with a LOF effect may be responsible for a cAMP reduction in nonresponding cases, while, conversely, GOF mutations could result in increased cAMP levels in responding patients. On this basis, a clear assessment of the direction of the alteration of cAMP (increase vs. decrease) resulting from different *GNAO1* mutations is crucial for establishing inclusion and exclusion criteria in clinical trials with caffeine in *GNAO1*-related dyskinesia.

In sum, our findings further confirm that *C. elegans* is a valid in vivo model for understanding the molecular and cellular mechanisms underlying GNAO1 encephalopathy. The generated strains represent an efficient platform to functionally classify the increasing number of *GNAO1* variants and perform broader pharmacological screens.

## Figures and Tables

**Figure 2 genes-14-00319-f002:**
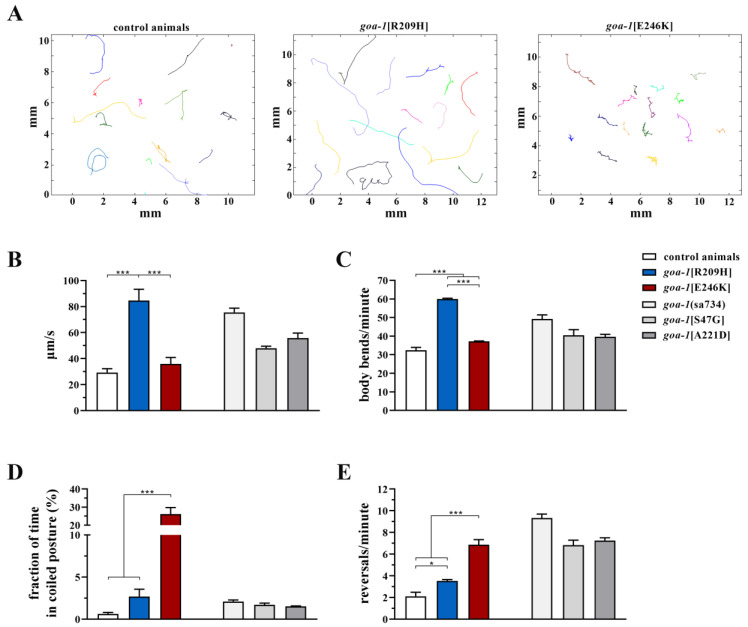
*goa-1*[R209H] and *goa-1*[E246K] animals exhibit aberrant motor behavior. (**A**) Trajectories of 15 worms on agar plates recorded for 2 minutes. Different colors refer to different animals. The gene-edited mutants showed abnormal crawling and longer (*goa-1*[R209H]) or irregular (*goa-1*[E246K]) tracks compared with the controls. (**B**) *goa-1*[R209H] crawled faster than wild-type animals (*** *p* < 0.0001; one-way ANOVA with Bonferroni correction). The aberrant motor behavior of both mutants was also revealed by the number of body bends per minute (**C**), the time spent in a coiled position (**D**), and the reversal rate (**E**) (* *p* < 0.05 and *** *p* < 0.0001). Twenty animals were tested for each genotype. Data represent means ± SEM of multiple observations. Data collected in our previous study [18] are included in panels B–E for comparison (gray bars). *sa734* is a null allele of *goa-1* [30,32].

**Figure 3 genes-14-00319-f003:**
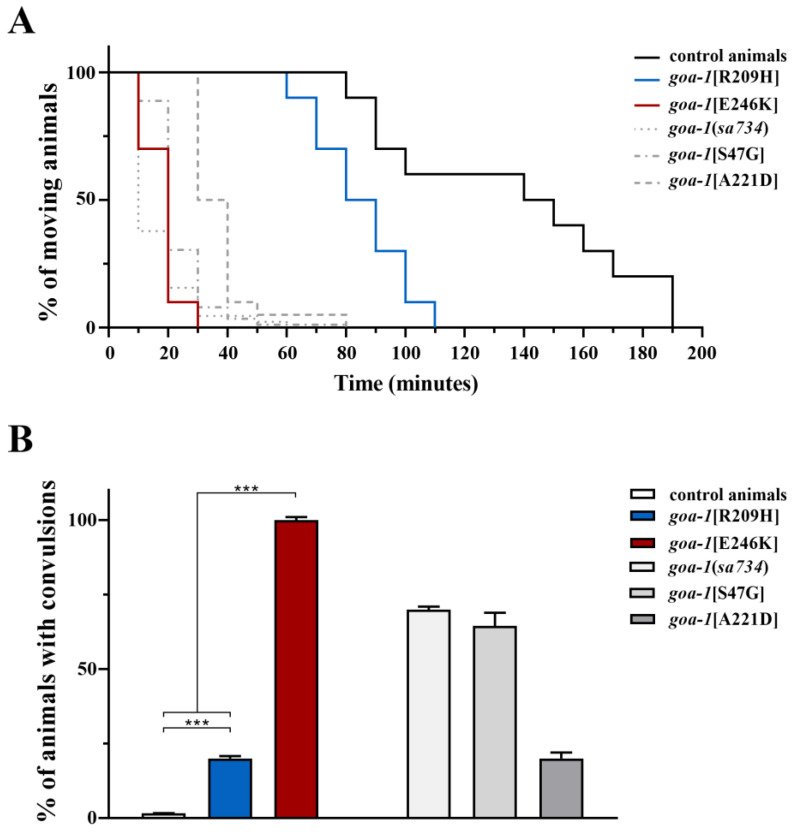
*goa-1*[R209H] and *goa-1*[E246K] animals display increased ACh release at the NMJ. (**A**) Gαo mutants showed hypersensitivity to aldicarb (1 mM) compared with control animals (*p* < 0.005 for *goa-1*[R209H] and *p* < 0.0001 for *goa-1*[E246K]; log-rank test), suggesting increased ACh release at the *C. elegans* NMJ. Twenty animals for each genotype were tested. (**B**) Gαo mutants showed hypersensitivity to PTZ (5 mg/mL on agar plates; 15 min of exposition), likely indicating an excess of stimulatory signal over inhibitory signal at the NMJ (*** *p* < 0.0001; Fisher’s exact test with Bonferroni correction). In both assays, *goa-1*[E246K] animals displayed a much more severe phenotype than *goa-1*[R209H] worms (*p* < 0.0001). Twenty animals for each genotype were tested. Data represent means ± SEM of three independent experiments. Data collected in our previous study [18] are included for comparison (gray bars). *Sa734* is a null allele of *goa-1* [30,32].

**Figure 4 genes-14-00319-f004:**
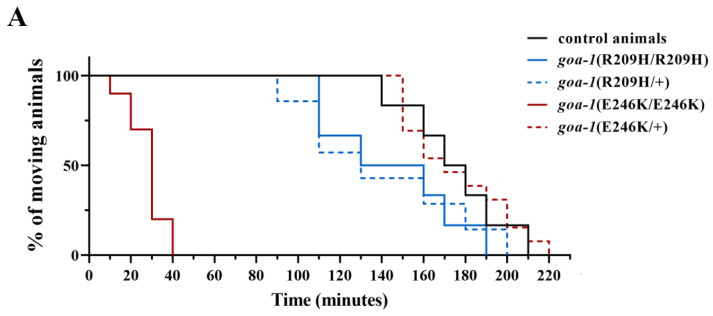
The monoallelic *goa-1*(+/R209H) allele displays a cell-specific dominant-negative behavior in *C. elegans* neurons. The sensitivity to aldicarb (**A**), the reversal rate (**B**), and the number of retained eggs (**C**) were assessed in worms carrying the *goa-1* mutations in a heterozygous state. *goa-1*(+/R209H) animals displayed hypersensitivity to aldicarb-induced paralysis (*p* < 0.005; log-rank test) and an increased reversal rate (* *p* < 0.05 and *** *p* < 0.0001; one-way ANOVA with Bonferroni correction), whose severity was undistinguishable compared with that of animals carrying the mutation in a homozygous state. These data suggest a DN behavior of the R209H variant in ventral cord motor neurons and possibly in AVA neurons, but not in HSN motor neurons (egg laying activity was not significantly affected in heterozygous worms) (**C**). No relevant phenotypes were observed in the hT2-balanced *goa-1*(+/E246K) mutants (PHX6121 strain). Twenty animals for each genotype were tested. Data represent means ± SEM of multiple experiments that were carried out using three unrelated clones generated following genetic crosses to control animals. Data collected in our previous study [18] are included in panels B and C for comparison (gray bars). *sa734* is a null allele of *goa-1* [30,32]. The behavior of *goa-1*(+/*sa734*), *goa-1*(+/S47G), and *goa-1*(+/A221D) heterozygous animals is also shown (panels **B**,**C**).

**Figure 5 genes-14-00319-f005:**
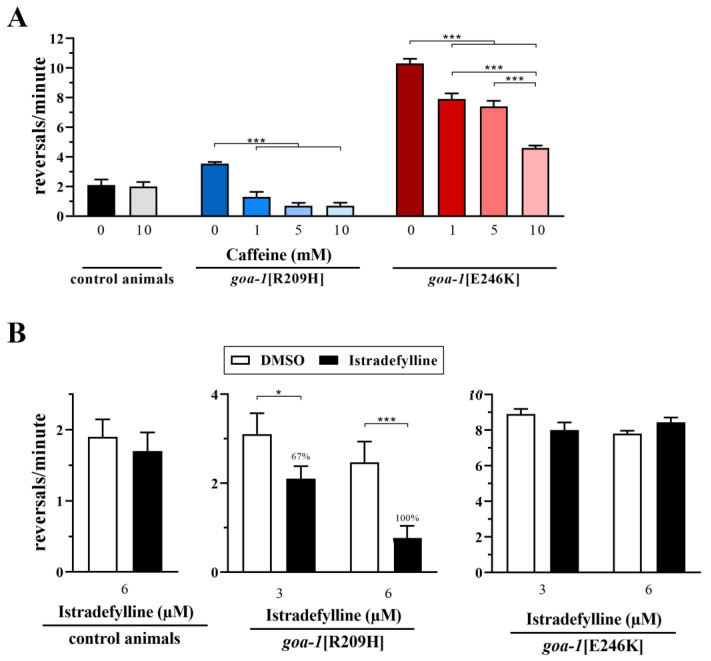
Caffeine and istradefylline variably improve the aberrant locomotor behavior of *goa-1* mutants. (**A**) Two hours of exposure to caffeine ameliorated the hyperactive locomotion of knock-in animals in terms of the number of short reversals per minute in a dose-dependent manner (*** *p* < 0.0001; one-way ANOVA with Bonferroni correction). (**B**) Exposure to istradefylline (2 h) decreased the reversal rate of *goa-1*[R209H] worms (* *p* < 0.05 and *** *p* < 0.0001) but not *goa-1*[E246K] animals, suggesting that caffeine also acts through an AR-independent mechanism. Results are also provided as the percentage of rescue (see Materials and Methods for details). Twenty animals for each genotype were tested. Data represent the means ± SEM of multiple observations.

**Table 1 genes-14-00319-t001:** Biallelic *goa-1* mutations exhibit a variable degree of developmental phenotypes.

Genotype	Embryonic Lethality (%)	Larval Arrest (%)	Brood Size (% of Controls)	Pvl Phenotype (%)
*goa-1*[WT]	0.5 ^1^	0	-	1
*goa-1*[R209H]	5 (*p* < 0.01) ^2^	5 (*p* < 0.01)	80 (*p* < 0.01)	10 (*p* < 0.01)
*goa-1*[E246K]	21 (*p* < 0.001)	40 (*p* < 0.001)	18 (*p* < 0.001)	50 (*p* < 0.001)

^1^ Twenty hermaphrodites per genotype were assayed to evaluate the brood size. More than 200 animals per genotype were tested in all other assays. ^2^ *p*-values are calculated using two-way ANOVA with Bonferroni correction.

## Data Availability

Not applicable.

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
