# Peer review of "Phenotypic Assessment of Pathogenic Variants in GNAO1 and Response to Caffeine in C. elegans Models of the Disease"

_genes, 2023, doi:10.3390/genes14020319_

Round 1
Reviewer 1 Report
Summary
Pathogenic mutations in Gao encoding gene (GNAO1) are causal to neurodevelopmental issues with childhood onset including movement disorders and epilepsy. The authors modeled several GNAO1 variants in C. elegans by generating goa-1 mutant worms. The authors report the effect of goa-1 mutations on previously established behavioral assessments. They report dominant negative and loss of function phenotypes for several variants, some of which have been previously described. The authors conclude with experiments suggesting therapeutic potential of caffeine and istradefylline on restoration of locomotor behavior. The data is significant in that there is no cure for GNAO1-related disorders and istradefylline is currently in clinical trials for Parkinson’s movement disease. The effect of caffeine is exciting as a proof of concept, however it is unclear how dosages would translate to mammalian systems. Interpretations would benefit from additional controls as mentioned below. Overall the manuscript is very well written and clear data organization.
Specific comments
1.) Examination of goa-1 knockout (and possibly goa-1 +/-) would provide helpful insightful as an additional control group.
2.) It is unclear why goa-1[R209H] performance in the aldicarb assay, compared with control, was substantially different between Figure 3A and Figure 4A. Line 272 is not accurate as the performance between goa1(R209H/R209H) and goa-1(R209H/+) was not similar. The R209H heterozygous animal exhibited significantly greater sensitivity. How do the authors reconcile such dominant negative effect?
3.) Figure 5 would benefit greatly from treatment of control animals with caffeine and istradefylline.
Author Response
We thank the reviewer for their overall positive evaluation and constructive comments.
1) Examination of goa-1 knockout (and possibly goa-1 +/-) would provide helpful insightful as an additional control group.
We thank the reviewer for this comment and apologize for the lack of clarity regarding this issue. Gαo null mutants have been extensively characterized and show hypersensitivity to aldicarb, hyperactive egg laying, and hyperactive locomotion [refs. 30,32-35]. Previous findings by our group and others established that knock-in animals carrying biallelic goa-1 variants phenocopy the behavior of goa-1 null mutants [refs. 17,18].
Here, to better comparing the phenotype of knock-in mutants with each other and with goa-1 deficient worms, we included our previous data in Figures 1-4 (gray bars). Similarly, we included data on goa-1(+/-) animals in Figure 4 (panels B, C). As for the aldicarb assay, Wang and colleagues have recently shown that goa-1(+/-) animals behave as control worms, displaying normal sensitivity to aldicarb-induced paralysis [ref. 17]. Conversely, goa-1(+/R209H) displayed aldicarb hypersensitivity, which was similar to that observed in homozygous goa-1[R209H] animals, suggesting a dominant-negative behavior for the R209H allele, at least for this phenotype. We revised the main text (Results section) to make these data more understandable (pages 8-9) and added a clarifying sentence in the legend of each figure of the revised manuscript.
2) It is unclear why goa-1[R209H] performance in the aldicarb assay, compared with control, was substantially different between Figure 3A and Figure 4A.
The reviewer is right on this point. Indeed, the aldicarb assay is known to be affected by a number of variables, including the age of the assay plates, the drug concentration, the temperature and humidity in the lab, the subjective nature of the method by which paralysis is scored, and the batch of the drug. During our experiments, we minimized this variability by performing blinded experiments under the same conditions and conducted by the same researcher. Unfortunately, we could not avoid variability due to the use of a different batch of aldicarb in the two experiments. As reported in the literature [ref. 24], comparisons between different mutants can only be performed in parallel on the same day using the same batch of aldicarb, while comparison of data from distinct assays is not always informative. In our experiments, goa-1[R209H] animals showed hypersensitivity to aldicarb in both Fig. 3A and Fig. 4A, although the absolute values were different. This discrepancy is likely due to the relative concentration of aldicarb, which appears to be slightly lower in the experiment shown in Fig. 4A compared to Fig. 3A.
3) Line 272 is not accurate as the performance between goa1(R209H/R209H) and goa-1(R209H/+) was not similar. The R209H heterozygous animal exhibited significantly greater sensitivity. How do the authors reconcile such dominant negative effect?
We thank the reviewer for identifying this error. In Fig. 4A, we have inaccurately used data from two different experiments. Based on the considerations reported above, it is not possible to compare data from aldicarb-sensitivity experiments performed at different times and with different dilutions/batches of aldicarb. We apologize for this mistake. The new Fig. 4A includes mean data obtained in parallel, within the same experiment.
4) Figure 5 would benefit greatly from treatment of control animals with caffeine and istradefylline.
Following the reviewer's suggestion, we tested the effect of caffeine (10 mM) and istradefylline (3 and 6 µM) in control animals. The results of these experiments, included in the new Fig. 5, indicate that both molecules have no effect on the reversal rate of wild-type worms, at least at the doses tested.

Reviewer 2 Report
The manuscript is a brief report on pathogenic GNAO1 variants and assesses the responses to caffeine and istradefylline in C. elegans diseased models. The study is concise but also informative. I recommend the manuscript for publication as brief report, subject to the required minor alterations listed below;
1- The manuscript title needs to be altered: "neurotransmitter homeostasis" is very broad and specifically was not analyzed in this study and was only interpreted and suggested indirectly by behavioral studies. Thus, the title should be more precise to respecting results obtained. Something similar to: "Behavioral/Phenotypic assessment of different pathogenic variants of GNAO1 and their respond to caffeine in C. elegans models of the disease"
Also, in my opinion, as it is now, the location of the word "differentially" is confusing, might be better to be altered to: "Pathogenic variants of GNAO1 respond differently/differentially to caffeine..."
2-English language needs spell checking. Example: Line 110, "...as the percentage of animals that still moves (move) at each time point."
3. Figures (specially bar charts) are too small to read the axis labels, titles and units. I recommend making them bigger and more readable.
Author Response
We thank the reviewer for their overall positive evaluation and constructive comments.
1) The manuscript title needs to be altered: "neurotransmitter homeostasis" is very broad and specifically was not analyzed in this study and was only interpreted and suggested indirectly by behavioral studies. Thus, the title should be more precise to respecting results obtained. Something similar to: "Behavioral/Phenotypic assessment of different pathogenic variants of GNAO1 and their respond to caffeine in C. elegans models of the disease". Also, in my opinion, as it is now, the location of the word "differentially" is confusing, might be better to be altered to: "Pathogenic variants of GNAO1 respond differently/differentially to caffeine...".
We changed the title to “Phenotypic assessment of pathogenic variants in GNAO1 and response to caffeine in C. elegans models of the disease” following the reviewer’s suggestion.
2) English language needs spell checking. Example: Line 110, "...as the percentage of animals that still moves (move) at each time point".
We thank the reviewer for catching this oversight, which was amended in the revised manuscript. We have also carefully revised the text as recommended.
3) Figures (specially bar charts) are too small to read the axis labels, titles and units. I recommend making them bigger and more readable.
We revised all Figures, including Suppl. Fig. 2, as recommended by the reviewer to make them more readable.

Round 2
Reviewer 1 Report
The authors have addressed my previous concerns. The additional control data makes it a high quality manuscript.